# Development of a method for measuring spleen stiffness by transient elastography using a new device and ultrasound-fusion method

Takaaki Tanaka◎, Masashi Hirooka◎, Yohei Koizumi◎, Takao Watanabe◎, Osamu Yoshida◎, Yoshio Tokumoto◎, Yoshiko Nakamura◎, Koutarou Sunago◎, Atsushi Yukimoto◎, Masanori Abe◎, Yoichi Hiasa ◎ *

Department of Gastroenterology and Metabology, Ehime University Graduate School of Medicine, Toon, Ehime, Japan

◎ These authors contributed equally to this work.
* hiasa@m.ehime-u.ac.jp

**Data Availability Statement:** All relevant data are within the manuscript and its Supporting information files.

## Abstract

### Background

Hepatic venous pressure gradient (HVPG) is the gold standard index for evaluating portal hypertension; however, measuring HVPG is invasive. Although transient elastography (TE) is the most common procedure for evaluating organ stiffness, accurate measurement of spleen stiffness (SS) is difficult. We developed a device to demonstrate the diagnostic precision of TE and suggest this technique as a valuable new method to measure SS.

### Methods

Of 292 consecutive patients enrolled in this single-centre, translational, cross-sectional study from June through September in 2019, 200 underwent SS measurement (SSM) using an M probe (training set, n = 130; inspection set, n = 70). We performed TE with B-mode imaging using an ultrasound-fusion method, printed new devices with a three-dimensional printer, and attached the magnetic position sensor to the convex and M probes. We evaluated the diagnostic precision of TE to evaluate the risk of esophagogastric varices (EGVs).

### Results

The median spleen volume was 245 mL (range, 64–1,720 mL), and it took 2 minutes to acquire a B-mode image using the ultrasound-fusion method. The median success rates of TE were 83.3% and 57.6% in patients with and without the new device, respectively (p<0.001); it was 76.9% and 35.0% in patients with and without splenomegaly (<100 mL), respectively (p<0.001). In the prediction of EGVs, the areas under the receiver operating characteristic curve were 0.921 and 0.858 in patients with and without the new device, respectively (p = 0.043). When the new device was attached, the positive and negative

**Funding:** This study was supported a part of funding from the Japan Society for the Promotion of Science (JSPS) KAKENHI Grant Number 18K07634 to MH, and 18K08007 to YH, and from AMED under Grant Number JP20fk0210058 to YH.

**Competing interests:** The authors have declared that no competing interests exist.

likelihood ratios were 3.44 and 0.11, respectively. The cut-off value of SSM was 46.0 kPa. Data that were similar between the validation and training sets were obtained.

## Conclusions

The SS can be precisely measured using this new device with TE and ultrasound-fusion method. Similarly, we can estimate the bleeding risk due to EGV using this method.

## Introduction

Liver cirrhosis is the final stage in the evolution of chronic liver disease, and the outcome of affected patients is regulated by the degree of portal hypertension (PH). PH is a clinical syndrome with hallmarks of portosystemic collaterals, splenomegaly, ascites, and encephalopathy [1]. Esophagogastric varices (EGVs) resulting from PH are a serious complication of cirrhosis. The estimated prevalence of EGVs in patients with cirrhosis has been reported to be 50% [2], and the mortality rate of variceal bleeding ranges from 20% to 35% [3]. To detect EGVs, screening endoscopy is recommended for all patients with cirrhosis [4,5]. However, a general program of routine endoscopic screening of these patients may be expensive and unnecessary. Therefore, non-invasive measures for diagnosing EGVs in cirrhotic patients before performing invasive screening endoscopy are required to avoid these challenges.

Over the past few decades, measurement of the hepatic venous pressure gradient (HVPG) has been established as a gold standard for the diagnosis and staging of PH [6–11]. It was previously demonstrated that an HVPG value higher than 10 mmHg predicts the presence of EGVs, whereas a value higher than 12 mmHg is predictive of variceal bleeding [12]. In contrast, HVPG measurement in routine clinical practice is limited by the invasiveness of the procedure and the technical expertise required for its performance. Consequently, different biomarkers and imaging modalities have been evaluated for their usefulness as non-invasive surrogate markers of PH [13–18].

Recently, Colecchia et al. showed that spleen stiffness (SS) is more accurate than other non-invasive parameters in identifying patients with EGVs and those with different degrees of PH [19,20]. Moreover, SS measurement (SSM) is useful as a non-invasive method to evaluate PH. We have previously proposed the splenic elasticity for PH (SEP) score (HVPG = splenic elasticity x 1.63–2.88) and reported a correlation between HVPG and SS [21]. However, the SEP score was measured using the real-time tissue elastography (RTE) method. Transient elastography (TE) is an easy and common method to evaluate organ stiffness. However, TE does not have B-mode imaging; hence, performing SSM is difficult and results in a high rate of inaccurate measurements. Thus, a new method of TE is needed for SSM. We developed a new device and printed on a three-dimensional (3D) printer, with TE and ultrasound-fusion method, to resolve this limitation. We aimed to assess the diagnostic performance of this method for SSM and determine its usefulness for estimating the risk of bleeding caused by EGVs.

## Materials and methods

### Study design

The study was approved by the Accredited Clinical Research Review Committee National University Corporation Ehime University Clinical Research Review Committee (IRB No. jRCTs062190005). The nature of the study was explained to the patients, and written informed

consent was obtained from each patient according to the principles of the Declaration of Helsinki. This was a single-centre, translational, cross-sectional study of patients admitted to the Department of Gastroenterology and Metabology, Ehime University Hospital from June 2019 to September 2019. Patients who underwent abdominal ultrasonography and whose liver stiffness and SS were measured using TE were recruited. All patients underwent computed tomography (CT) scans from the upper abdomen to the pelvic cavity. The selection criteria were: 1) 20 to 90 years of age and 2) Eastern Cooperative Oncology Group (ECOG) Performance Status of 0 to 2. Notably, the patients' sex was not considered in the course of selection. Abdominal ultrasonography was performed in the laboratory of the inpatient ward; on the same day, patients underwent B-mode examination and measurements using TE on an empty stomach more than 2 hours after eating. All participants were provided with a written explanation of the purpose, method, and details regarding handling personal information of the research.

The primary endpoint was set by comparing the test success rates. We compared the "success rate of the conventional method without attachments and that of the new method with attachments" using the paired t-test. When the change was 0.06, the standard deviation of the conventional and new methods was 0.4, respectively, and the correlation coefficients were set to 0.75, α 0.05, and 1-β 0.8. Although there were 177 cases, the total number of cases was 200, considering those who dropped out. The primary endpoint was the measurement success rate, which was defined based on the following criteria: 1) estimation of SS until 10 measurement values are obtained; 2) estimation of SS 20 times in an event 10 measurement values are not obtained. The measurement rate was calculated as: The number of times the measured value was obtained / the number of measurements. The secondary endpoints were examination time, measured interquartile range (IQR)/ median, and diagnostic ability for EGV. The analysis of the primary and secondary endpoints was based on the Full Analysis Set (FAS). Similarly, we conducted an analysis targeting the Per Protocol Set (PPS) and confirmed the stability of the analysis results. Values of $p < 0.05$ were considered statistically significant.

SSM was performed in 292 patients with chronic liver disease between June 2019 and September 2019. Exclusion criteria were as follows: 1) pregnancy in females, 2) previous splenectomy, and 3) previous open abdominal surgery. Ninety-two patients were excluded from this study because TE image could not be obtained by the M probe due to severe obesity. The remaining 200 enrolled patients were classified into two groups (training and validation sets). Consecutive patients enrolled from June 2019 to August 2019 constituted the training set to evaluate the diagnostic accuracy of TE for EGVs and determine the cut-off level for screening for high-risk EGVs. In the training set, valid SSM could not be obtained in 28 (21.5%) of the 130 patients who were initially evaluated due to unreliable results (success rate <60%; IQR/ median >30%), leaving a total of 102 patients. Subsequently, to validate the diagnostic accuracy of TE for SSM, 70 patients were enrolled between August 2019 and September 2019 and included in the validation set. Twenty patients (28.6%) were excluded due to unreliable results, leaving a total of 50 patients. Clinical and laboratory data were collected just before TE.

### TE with virtual B-mode imaging using the ultrasound-fusion method

Spleen and liver stiffness were measured using vibration-controlled transient elastography (VCTE) with an M probe from FibroScan502 (EchoSens, Paris, France), by two operators (T. T. and Y.K., with 3 and 10 years of experience, respectively). To confirm the reproducibility, only 20 of 200 patients were measured by both T.T and Y.K. The remaining 180 patients were measured by T.T. only. The liver stiffness measurement was obtained as in previous reports [22]. The criteria for reliability were 10 validated measurements and a median (IQR) success rate of at least 60% or <0.3 of the obtained liver and spleen stiffness values. For assessing the

**A**

**B**

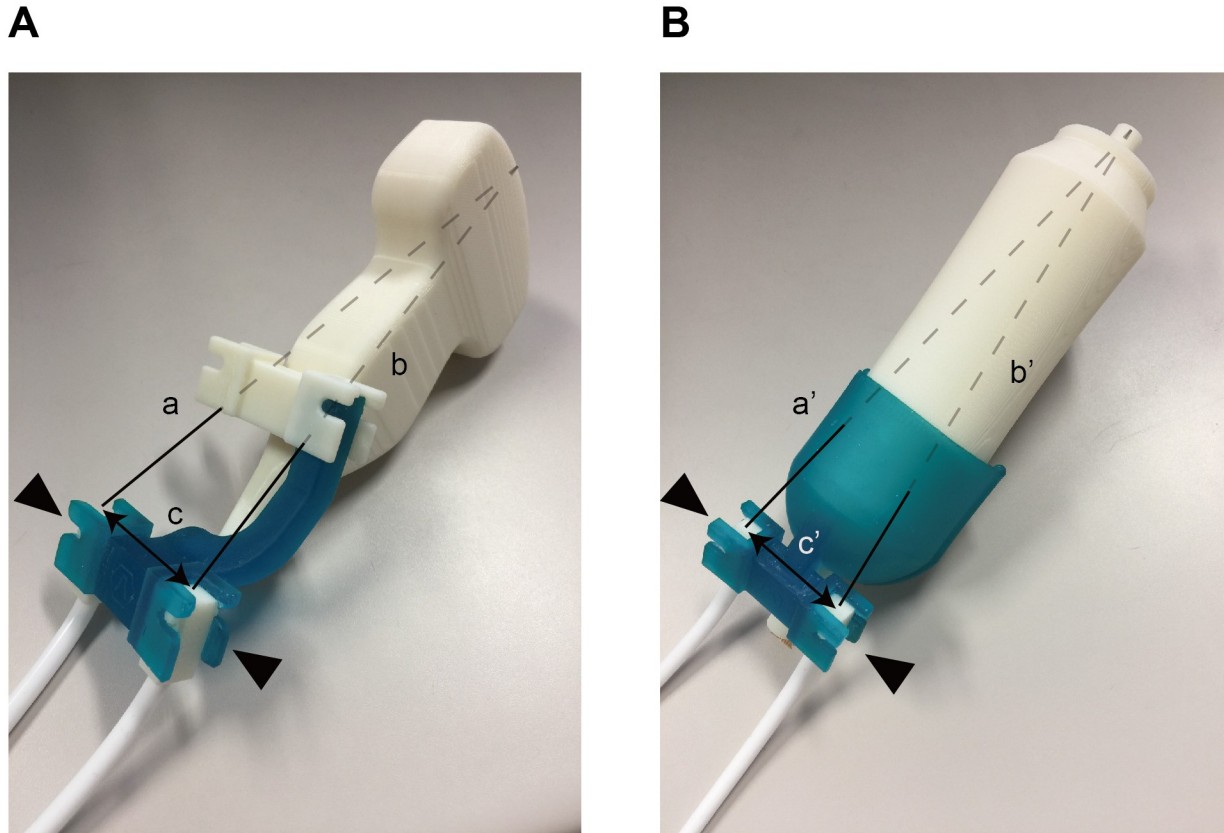

**Fig 1. New devices to attach to the ultrasound probes.** A) To attach the electromagnetic position sensors (black arrowhead) with the convex probe, we developed a device using the three-dimensional (3D) printer. B) To attach the sensors to the M probe, another new device (black arrowhead) was similarly developed. Each distance was matched between the tip of the probe and the position sensors (a = a', b = b', c = c').

SSM, the patient was placed in the supine position with the left arm in maximum abduction, and the transducer was placed in the left intercostal spaces. To confirm the B-mode images, we followed the ultrasound-fusion method [23,24], which was performed with TE with the newly developed device (Fig 1). Although two different types of devices were used to adjust the measurements, for the attached M and convex probes, each distance was matched between the tip of the probe and the position sensors (a = a'; b = b'; and c = c'). First, the new device with the electromagnetic position sensors was attached to the convex probe (4.0-MHz curvilinear C1-6, GE Healthcare) (S1 Movie). By tilting the convex probe with both the new devices and the sensors, 3D ultrasound volume data were acquired (S2 Movie). An ultrasound machine (LOGIQ E9, GE Healthcare, Chalfont St. Giles, UK) and a low magnetic field generator were used. Both the low magnetic field generator and position sensors were connected to a position-sensing unit embedded in the ultrasound machine. The 3D ultrasound volume data contained spatial information in the generated magnetic field. Next, the sensors were removed from the new device with the convex probe and attached to the other new device with an M probe (Fig 1B) (S1 Movie). Fusion imaging revealed the multiplanar reconstruction (MPR)-ultrasound images, which were constructed from the 3D ultrasound volume data by the direction of the M probe (Fig 2) (S2 Movie). SSM was performed while the ultrasound-fusion image was displayed. Two dimensional (2D)-shear wave elastography (SWE) was performed with the same ultrasound machine.

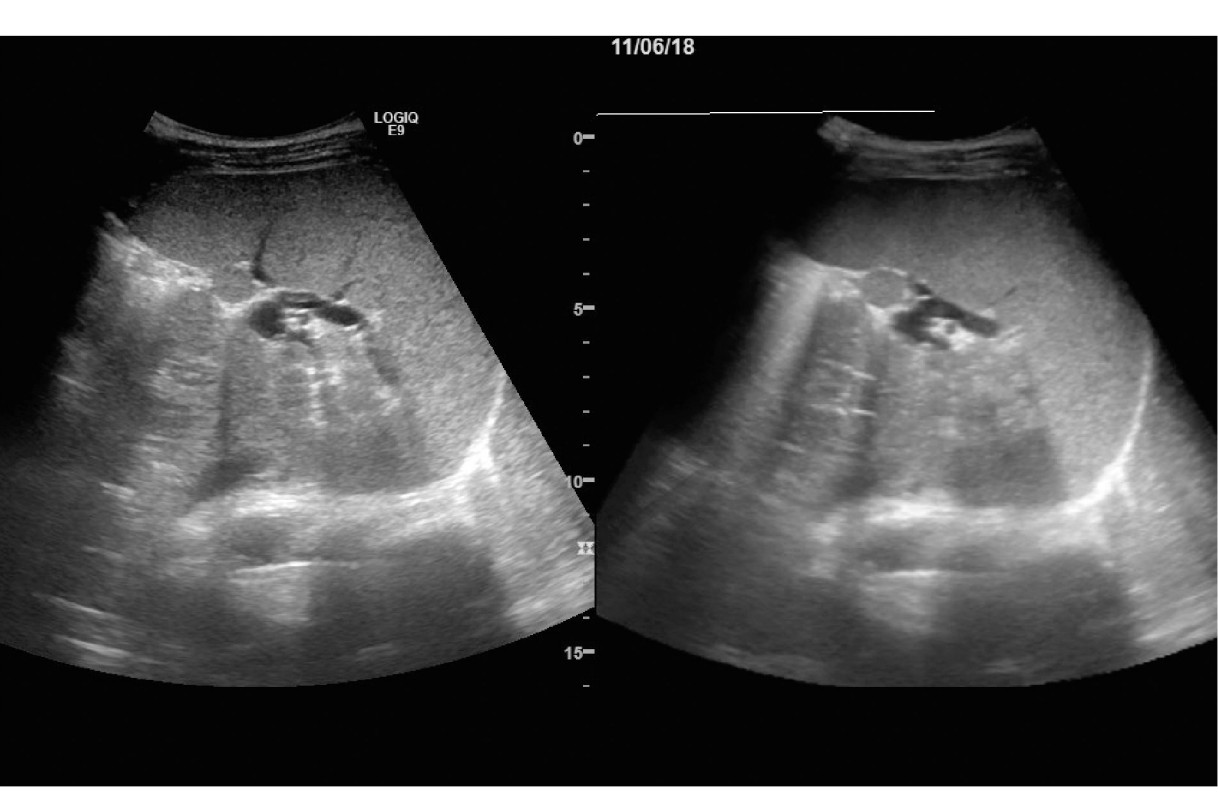

**Fig 2. B-mode image of the spleen constructed by ultrasound-fusion image.** The ultrasound-fusion image shows the synchronized view from the real-time B-mode imaging (A) and multiplanar reconstruction (MPR)-ultrasound images (B: Past B-mode imaging, so-called virtual B-mode imaging) simultaneously, side-by-side, in the display of the ultrasound device.

### Esophagogastroduodenoscopy and hepatic venous pressure gradient

Two independent, experienced endoscopists performed the examinations with good agreement to evaluate the size of EGVs (κ-value, 0.95). If discrepancy occurred, final reports, based on a consensus, were adopted. Endoscopic findings were recorded according to the following criteria: F1: Linear dilation, meandering varices; varices were flattened by insufflation. F2: Varices-like prayer beads; varices were not flattened by insufflation. F3: Nodular or mass-like thick varices; varices were not flattened by insufflation. Similarly, whether it was accompanied by a red colour (RC) sign was recorded. Varices with a high risk of rupture were identified as F1 with positive RC sign, or F2, or higher.

For measuring HVPG, the right hepatic vein was catheterized through the right femoral vein, and pressure in both the wedged and free positions was measured using a 5-Fr balloon-tipped catheter. The HVPG was calculated by subtracting the free hepatic venous pressure from the wedged venous pressure.

### Statistical analysis

All statistical analyses were performed using SAS version 9.4 (SAS Institute, Cary, NC). Data are presented as medians and IQR. The Kruskal-Wallis test was used with nonparametric data. To predict EGVs, receiver operating characteristic (ROC) curve analysis was performed. The

area under the ROC curve (AUC) was calculated by the trapezoidal rule. Optimal cut-off values for predicting EGVs were selected to maximize the sensitivity, specificity, and accuracy. Spearman's rank correlation coefficient was used to evaluate the correlation between SSM and HVPG. The prediction of the EGVs was calculated, and the performance of SSM for the evaluation of the training set compared with that of validation set was set as the gold standard using a κ agreement test.

## Results

### The reliability of SSM and patient characteristics

After excluding the patients based on the XL probe measurements, the median (IQR) success rates were compared with and without using the new device (n = 200, Fig 3). The median value of SSM was not significantly different between the 2 researchers (Y.K and T.T.) (p = 0.899). The median SSM success rate was 83.3% (55.6–100%) with the new device and 57.6% (0–83.3%; p <0.001) without the new device (Fig 4A). Especially in patients without splenomegaly (<100 mL), the median TE success rates were 76.9% and 35.0% with and without the new device, respectively (p <0.001). The median value (IQR) of SSM was 0.19 (0.10–0.30) and 0.21 (0.06–0.31; p = 0.785) with and without the new device, respectively (Fig 4B).

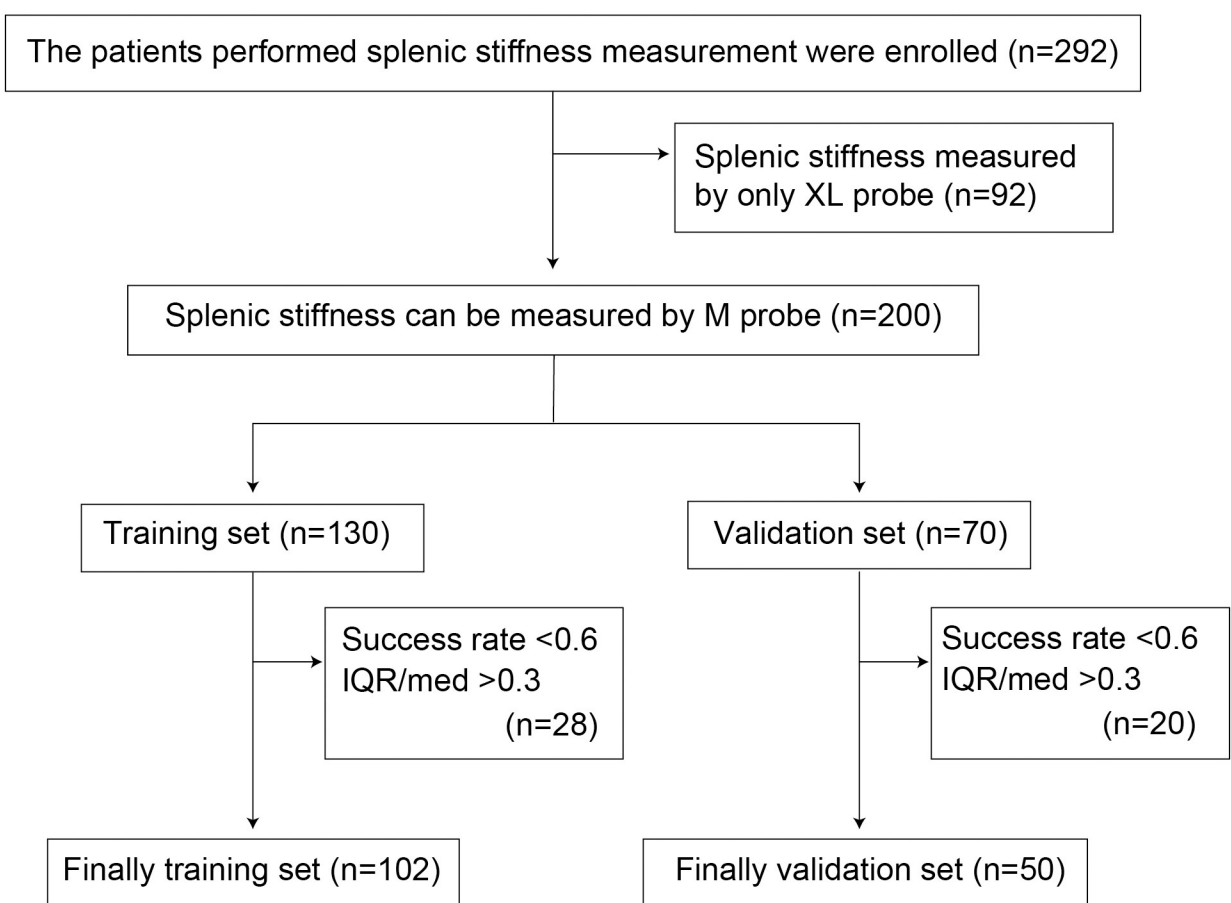

**Fig 3. Clinical profiles of patients for the spleen stiffness measurement with and without the new device.** IQR = interquartile range.

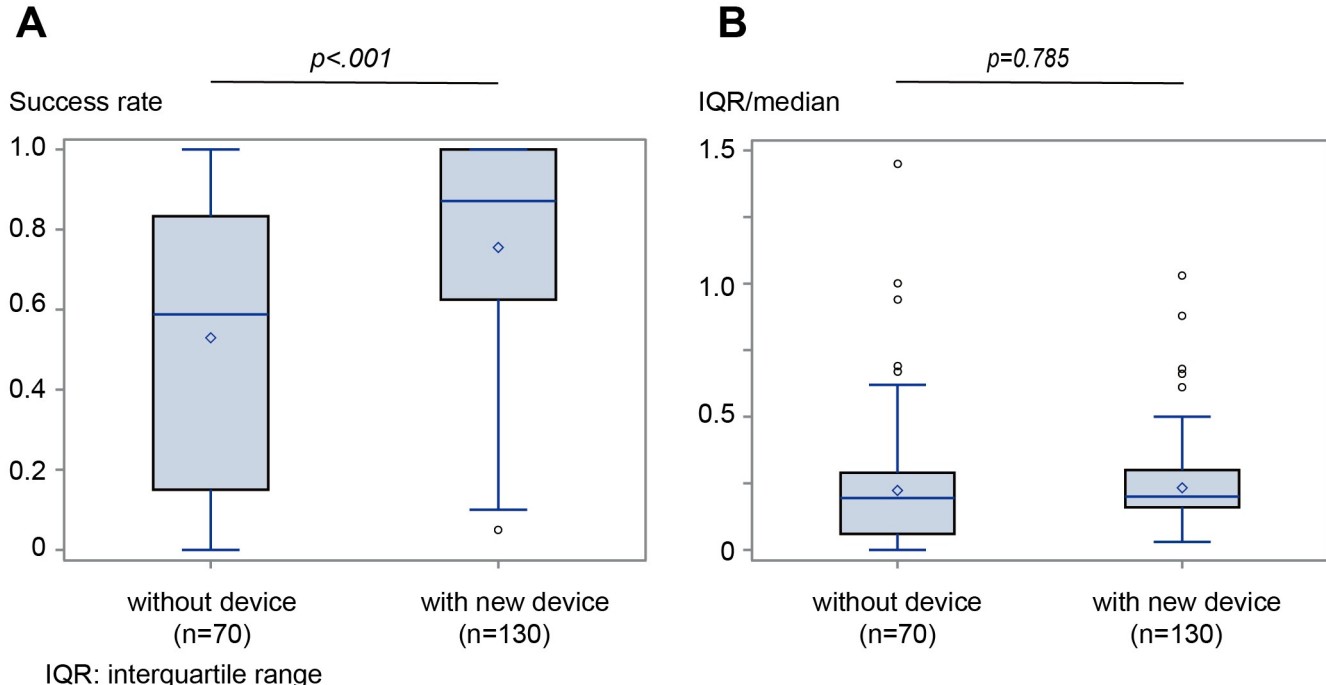

**Fig 4. Median success rate and IQR/median of SSM with and without the new device.** A) Median success rate of the SSM with the new device was higher than that of the SSM without one. B) No significant difference is observed between SSM with and without the new device. IQR = interquartile range, SSM = spleen stiffness measurement.

Moreover, the characteristics of the two independent study groups (training and validation sets) after adequate SSM in the enrolled patients are shown in Table 1. Age, sex, etiology, platelet counts, prothrombin time (PT), total bilirubin, fasting plasma glucose, and haemoglobin A1c were not significantly different between the training and validation set.

**Table 1. Clinical characteristics and laboratory data of patients.**

| | Training set (n = 102) | Validation set (n = 50) | p value |
|---|---|---|---|
| Age (years) | 69 (59–75) | 71 (59–72) | 0.572 |
| Male: female | 74: 28 | 36: 14 | 0.943 |
| Aetiology (HBV: HCV: NBNC: Non-liver disease) | 12: 29: 44: 17 | 2: 17: 25: 6 | 0.338 |
| Platelet counts | 13.3 (9.6–17.8) | 13.2 (8.2–14.8) | 0.842 |
| PT (%) | 84.8 (64.1–101.7) | 81.3 (61.1–98.3) | 0.571 |
| T. Bil | 0.9 (0.7–1.6) | 0.9 (0.6–1.4) | 0.473 |
| ALT | 34 (21–62) | 24 (15–45) | 0.076 |
| Albumin | 3.7 (3.2–4.3) | 3.7 (2.9–4.1) | 0.643 |
| FIB-4 index | 3.91 (2.59–6.32) | 3.71 (2.00–6.73) | 0.783 |
| M2BPGi | 2.21 (1.02–7.08) | 2.49 (1.08–5.8) | 0.855 |
| Splenic volume | 245 (136–406) | 216 (162–374) | 0.776 |

HBV = hepatitis B virus s antigen positive, HCV = anti hepatitis C virus positive, NBNC = both HBs antigen and anti HCV negative, PT = Prothrombin time, T. Bil = total bilirubin, ALT = alanine aminotransferase, FIB-4 (Fibrosis-4) was calculated as follows = age (years) × AST (IU/L)]/[PLT (× 109/L)] × [$\sqrt{}$ALT (IU/L)], M2BPGi = Mac-2 binding protein.

## Diagnostic accuracy for predicting EGVs and complication of PH in the training set

Diagnostic accuracy in predicting high-risk varices in the training set are shown in Fig 5A and 5B and Table 2. SSM with the new device had higher AUC (AUC, 0.880; 95% confidence interval (CI): 0.714–0.917; P <0.0001) than with other methods. The diagnostic accuracy of EGVs analyzed using the corresponding SSM with the new device from the ROC curves corresponded to the cut-off values of SSM at 46.4. The diagnostic accuracy of the cut-off values for SSM with the new device in the validation set had sensitivity, specificity, positive predictive value (PPV), negative predictive value (NPV), positive likelihood ratio (LR+), and negative likelihood ratio (LR-) of 92.3%; 72.4%; 53.3%; 74.5%; 3.34; and 0.11, respectively. Both SSM with and without the device had low LR-. Regarding other factors, the cut-off values of platelet and liver stiffness measurement were 119,000 /mm$^3$ and 20.7 kPa, respectively. The diagnostic accuracy of complications of PH (presence of ascites and/or encephalopathy) was similar to that of EVs (Fig 5C and 5D and Table 3). SSM with the new device had higher AUC (AUC, 0.887; P <0.0001) than with other methods.

A previous study reported that platelet count (150,000/mm3), liver stiffness measurement (LSM) (20 kpa), and SSM (46 kpa) were useful factors for the screening of EGVs [25]. When the platelet count changed from 150,000/mm$^3$ to 120,000/mm$^3$, the kappa coefficient improved (Table 4). Although both SSM with and without the new devices were linearly correlated with HVPG, the r-value was higher for SSM with the new device than without the new device (Fig 6).

## Discussion

The SSM success rate was remarkably increased when the B-mode image for the spleen was confirmed using the newly developed device. Splenic size and configuration varied with typical parameters commonly quoted for the normal splenic size of 12 × 7 × 4 cm and weight of 150 g (range 100–200 g).

The spleen is an organ in the abdominal cavity with a smooth serosal surface, which is attached to the retroperitoneum by a falciform-fat ligament with blood vessel supply. The splenic surface is described in conjunction with anatomical positions, including the phrenic nerve innervating the diaphragm and some internal organs' surface. The internal organ surface is classified into the upheaval of a front part or the stomach and the rear part or the kidney. Thus, visualization of the spleen by ultrasound is difficult, while that of the liver is easy. The rates of SSM failure were similar in two previous studies (14% vs. 11%) [19,20]. This was because of the high body mass index (BMI) in some patients; however, this was more frequently because of the success rate of measurement, which was <60%. SSM failed less frequently in patients with larger measurable spleen volume. In fact, it was easier to localize, under ultrasound guidance, the middle of the region of interest (ROI) in a spleen with a longitudinal diameter of at least 12 cm and a mean anteroposterior diameter of 4 cm [26].

Recently, RTE and acoustic radiation force impulse (ARFI) imaging are suggested as new alternative methods to evaluate SS. We can integrate the imaging modality with real-time B mode imaging. A promising result regarding the precision of the RTE and ARFI technology for non-invasive assessments of organization-induced fibrosis has been reported [21,27]. Because measurement by TE is based on M- and A-mode imaging, it is influenced by the experience of the operator. RTE and the ARFI imaging can be performed under the clear observation of the real measurement position by B-mode imaging; therefore, this can be applied in obese patients. The important clinical advantage of ARFI imaging is that it has a higher success rate of elastic measurement than TE [28,29]. Thus, we developed a new method of TE with B-

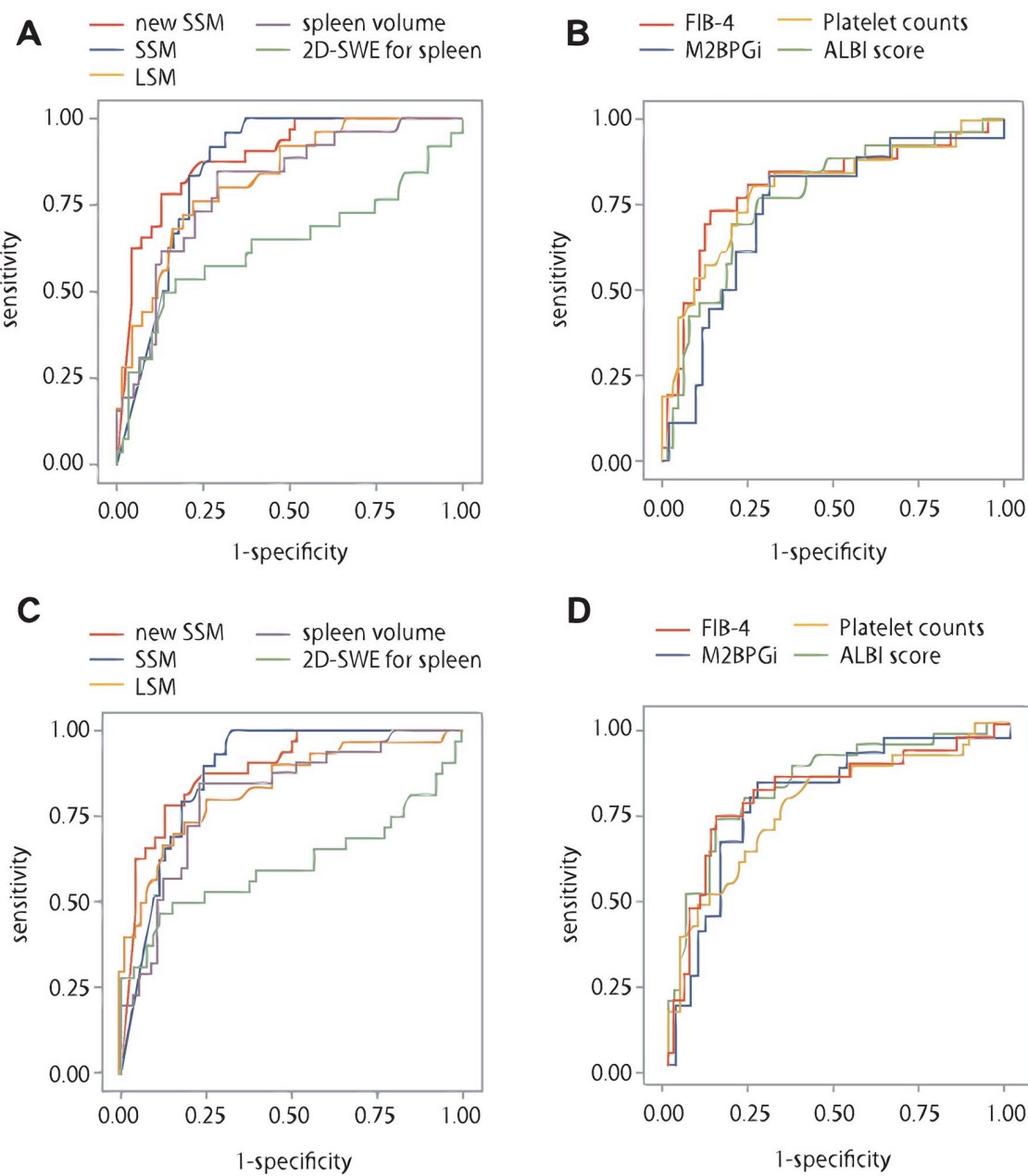

SSM: spleen stiffness measurement
LSM: liver stiffness measurement
SWE: shear wave elastography
M2BPGi: Mac-2 binding protein
ALBI: alubumin-bilirubin score

**Fig 5. Diagnostic accuracy for predicting EGVs and complications of portal hypertension in the training set.** A) ROC curves for the diagnostic accuracy for predicting EGVs by imaging. B) ROC curves for the diagnostic accuracy for predicting EGVs by biochemical markers. C) ROC curves for the diagnostic accuracy for predicting complications of PH by imaging. D) ROC curves for the diagnostic accuracy for predicting complications of portal hypertension by biochemical markers. SSM = spleen stiffness measurement, LSM = liver stiffness measurement, SWE = shear wave elastography, M2BPGi = Mac-2 binding protein, ALBI = albumin-bilirubin score, ROC = receiver operating characteristic, EGVs = esophagogastric varices, PH = portal hypertension.

**Table 2. Diagnostic accuracy for predicting high-risk varices in the training set.**

| Training set (n = 102) | Cut-off | Sensitivity | Specificity | PPV | NPV | LR+ | LR− | AUC |
|---|---|---|---|---|---|---|---|---|
| New SSM | 46.4 kpa | 92.3 | 72.4 | 53.3 | 74.5 | 3.34 | 0.11 | 0.880 |
| SSM | 45.0 | 91.7 | 73.1 | 55.0 | 73.6 | 3.41 | 0.11 | 0.858 |
| LSM | 20.7 | 76.0 | 77.9 | 55.9 | 73.1 | 3.44 | 0.31 | 0.827 |
| Spleen volume | 260 | 84.6 | 71.0 | 55.0 | 70.5 | 2.91 | 0.22 | 0.805 |
| 2D-SWE | 2.34 | 53.8 | 83.1 | 58.3 | 69.4 | 3.18 | 0.56 | 0.640 |
| FIB-4 | 5.84 | 73.1 | 85.9 | 67.9 | 71.1 | 5.20 | 0.31 | 0.805 |
| Platelet | 11.9 | 80.8 | 73.4 | 55.3 | 71.1 | 3.04 | 0.26 | 0.799 |
| M2BPGi | 2.69 | 83.3 | 68.6 | 48.4 | 73.9 | 2.66 | 0.24 | 0.737 |
| ALBI | -2.05 | 69.2 | 79.6 | 58.1 | 71.1 | 3.41 | 0.39 | 0.768 |

New SSM = spleen stiffness measurement with new device, SSM = spleen stiffness measurement without new device, LSM = liver stiffness measurement, 2D-SWE = two dimension-shear wave elastography, FIB-4 was calculated as follows = age (years) × AST (IU/L)]/[PLT (× 109/L)] × [$\sqrt{}$ALT (IU/L)], M2BPGi = Mac-2 binding protein, ALBI = albumin-bilirubin score, PPV = positive predictive value, NPV = negative predictive value, LR+ = positive likelihood ratio, LR- = negative likelihood ratio, AUC = Area under the ROC curve.

mode imaging. In an environment with a B-mode image signal by an ultrasound machine other than TE, tissue stiffness measurement cannot be performed. For this reason, TE cannot be simultaneously performed with real-time B-mode imaging. The ultrasound-fusion image shows the synchronized view of the simultaneous real-time B-mode imaging and MPR-ultrasound images (past B-mode imaging, so-called virtual B-mode imaging) side-by-side in the display of the ultrasound device (Fig 3B). In a low magnetic field, the MPR-ultrasound plane, defined by electromagnetic position sensors, was similarly reconstructed as the plane in real-time B-mode imaging. Thus, if the 3D ultrasound volume containing spatial information in the generated magnetic field was acquired by scanning in a manual sweeping manner and electromagnetic position sensors are attached to the M or XL probes, ultrasound-fusion imaging could be used in TE. To attach the electromagnetic position sensors, new devices were developed using a 3D printer. Even when using the new device, whether the push pulse is adequately generated to the spleen is unknown. However, the operator, assisted by a time-motion image,

**Table 3. Diagnostic accuracy for predicting the complication of portal hypertension in the training set.**

| | Cut-off | Sensitivity | Specificity | PPV | NPV | LR+ | LR− | AUC |
|---|---|---|---|---|---|---|---|---|
| New SSM | 57.1 | 78.1 | 87.1 | 73.5 | 68.6 | 6.08 | 0.25 | 0.887 |
| SSM | 54.2 | 72.4 | 82.2 | 65.6 | 68.1 | 4.08 | 0.34 | 0.880 |
| LSM | 23.7 | 76.0 | 77.9 | 55.9 | 73.1 | 3.46 | 0.31 | 0.834 |
| Spleen volume | 20.6 | 80.0 | 77.0 | 63.2 | 67.0 | 3.49 | 0.26 | 0.808 |
| 2D-SWE | 2.58 | 46.9 | 88.7 | 71.4 | 62.4 | 4.14 | 0.60 | 0.611 |
| FIB-4 | 5.84 | 65.6 | 87.9 | 75.0 | 64.4 | 5.44 | 0.39 | 0.784 |
| Platelet | 12.9 | 78.1 | 65.5 | 55.6 | 64.4 | 2.27 | 0.34 | 0.766 |
| M2BPGi | 2.69 | 82.6 | 73.9 | 61.3 | 66.7 | 3.17 | 0.24 | 0.785 |
| ALBI | -2.05 | 71.9 | 86.2 | 74.2 | 64.4 | 5.21 | 0.32 | 0.829 |

New SSM = spleen stiffness measurement with new device, SSM = spleen stiffness measurement without new device, LSM = liver stiffness measurement, 2D-SWE = two dimension-shear wave elastography, FIB-4 was calculated as follows = age (years) × AST (IU/L)]/[PLT (× 109/L)] × [$\sqrt{}$ALT (IU/L)], M2BPGi = Mac-2 binding protein, ALBI = albumin-bilirubin score, PPV = positive predictive value, NPV = negative predictive value, LR+ = positive likelihood ratio, LR- = negative likelihood ratio, AUC = Area under the ROC curve.

**Table 4. Screening for high-risk esophago-gastric varices.**

| | Training set (n = 102) | | Validation set (n = 50) | |
|---|---|---|---|---|
| | High-risk varices | | High-risk varices | |
| | + | - | + | - |
| Plt>15, LSM<20, SSM<46 | 0 | 33 | 0 | 14 |
| Plt≤15, LSM≥20, SSM≥46 | 26 | 43 | 11 | 25 |
| *Kappa coefficient* | *0.281* | | *0.198* | |
| Plt>12, LSM<24, SSM<46 | 0 | 44 | 0 | 20 |
| Plt≤12, LSM≥24, SSM≥46 | 26 | 32 | 11 | 19 |
| *Kappa coefficient* | *0.412* | | *0.317* | |

Plt>15 = platelet count > 150,000/μL, LSM<20 = liver stiffness measurement <20 kPa, SSM<46 = spleen stiffness measurement <46 kPa.

can locate the spleen portion at least 6 cm deep and free of large vascular structures. Thus, this establishes the case for acquired SSM, as the spleen is the only organ in the left hypochondrium whose stiffness can be measured by TE. Thus, the spleen can be measured appropriately from the acquired SSM numerical values. SSM was mainly measured by T.T. (with only 3 years of TE experience). There was a high reproducibility between the non-expert and the expert. To detect the high-risk EGV group, the cut-off value of SSM was reported as 46 kPa in a previous report [24]. In this study, a cut-off level similar to that in the previous report was adopted.

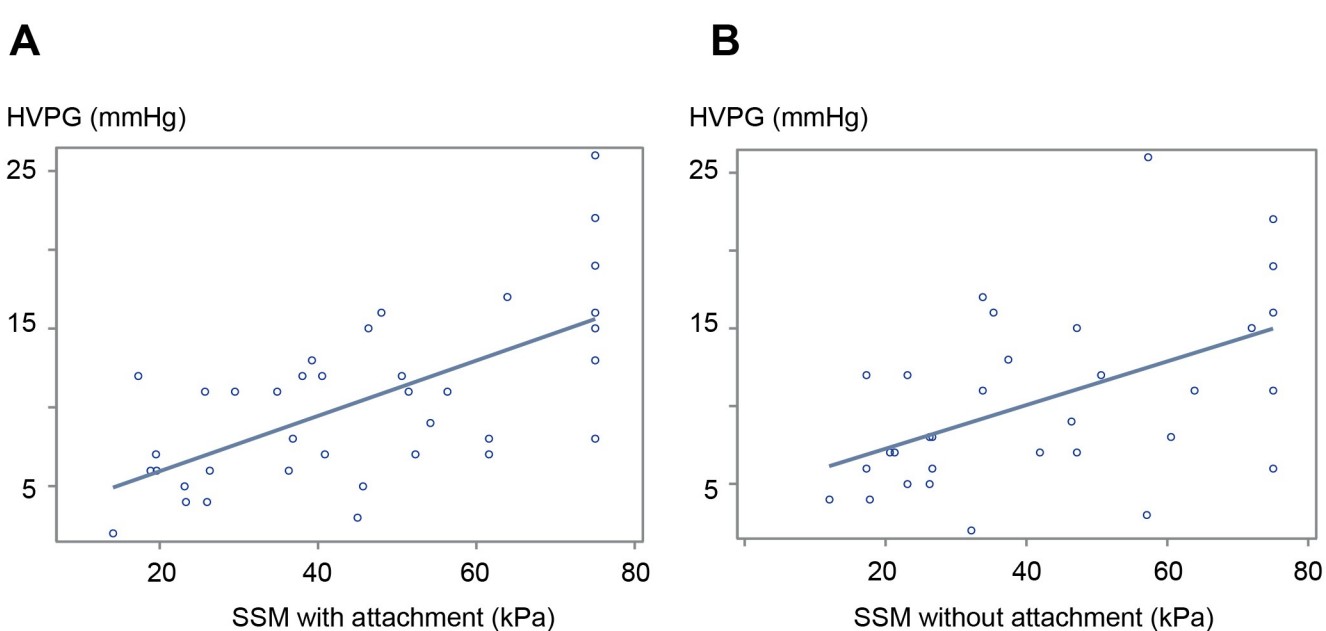

HVPG: hepatic venous pressure gradient
SSM: spleen stiffness measurement

**Fig 6. SSM with and without the new device linearly correlated with HVPG.** A) SSM with the new device, B) SSM without the new device. SSM = spleen stiffness measurement, HPVG = hepatic venous pressure gradient.

SSM with the new device is a better predictive marker of patients in the high-risk EGV, ascites, or hepatic encephalopathy group. However, SSM without the new device was a better marker when the SSM value could be acquired. Furthermore, the result of 2D-SWE was unexpected; however, the reason could not be clarified. The coloration heterogeneity within the ROI was evident compared to the measurement in the liver; therefore, the 2D-SWE may not have been measured properly. In this study, the cut-off value of the platelet count was 120,000/mm$^3$; thus, the reason for the difference in etiology.

This study had some limitations. First, data collection was performed in a single center, primarily to confirm the efficacy of the newly developed device. To validate the clinical efficacy, multicenter studies need to be performed. Secondly, unlike the liver, histological diagnosis cannot be easily performed in the spleen. Thirdly, this was a cross-sectional study; thus, a prospective study should be performed.

In conclusion, SSM can be precisely performed using this new device with TE and the ultrasound-fusion method. Moreover, the risk of bleeding caused by EGV was estimated by the method.

## Supporting information

**S1 Movie. Setting a self-made device and replacement.** A self-made device is set to the convex probe, and a position sensor is set to the device. At this time, a mark is made on the right side to prevent a mix up between the left and right sides of the position sensor. (A) Removal of the position sensor from the convex probe device. (B) Setting the self-made device to the M probe during TE. A position sensor is set to the device. At this time, the marking is made on the right side.
(ZIP)

**S2 Movie. Ultrasound-fusion image creation and spleen stiffness measurement.** A). A convex probe is used to visualize the spleen on the left side. At this time, the splenic hilar portion is drawn with the maximum diameter. An ultrasound-fusion image is created by sliding the convex probe from head to tail between the same ribs. (B) The spleen stiffness is measured on the same ribs by TE, a self-made device, and position sensor. The Logic E9 monitor is split into two images, with the ultrasound-fusion image on the right. By vibrating the fusion image, the possibility of measuring the stiffness by TE is determined. TE = transient elastography.
(ZIP)

**S1 File.**
(XLSX)

**S2 File.**
(XLSX)

## Author Contributions

**Conceptualization:** Masashi Hirooka, Yoichi Hiasa.

**Data curation:** Takaaki Tanaka, Masashi Hirooka, Yohei Koizumi, Takao Watanabe, Osamu Yoshida, Yoshio Tokumoto, Yoshiko Nakamura, Koutarou Sunago, Atsushi Yukimoto.

**Formal analysis:** Takaaki Tanaka, Masashi Hirooka, Yohei Koizumi, Takao Watanabe, Osamu Yoshida, Yoshio Tokumoto, Yoshiko Nakamura, Koutarou Sunago, Atsushi Yukimoto.

**Funding acquisition:** Masashi Hirooka, Yoichi Hiasa.

**Investigation:** Takaaki Tanaka, Yohei Koizumi.

**Methodology:** Masashi Hirooka.

**Project administration:** Masashi Hirooka, Yoichi Hiasa.

**Resources:** Takaaki Tanaka, Masashi Hirooka, Yohei Koizumi, Takao Watanabe, Osamu Yoshida, Yoshio Tokumoto, Yoshiko Nakamura, Koutarou Sunago, Atsushi Yukimoto.

**Software:** Takaaki Tanaka, Masashi Hirooka, Yohei Koizumi, Takao Watanabe, Osamu Yoshida, Yoshio Tokumoto, Yoshiko Nakamura, Koutarou Sunago, Atsushi Yukimoto.

**Validation:** Takaaki Tanaka, Yohei Koizumi.

**Visualization:** Takaaki Tanaka, Masashi Hirooka.

**Writing – original draft:** Takaaki Tanaka, Masashi Hirooka, Yoichi Hiasa.

**Writing – review & editing:** Takaaki Tanaka, Masashi Hirooka, Yohei Koizumi, Takao Watanabe, Osamu Yoshida, Yoshio Tokumoto, Yoshiko Nakamura, Koutarou Sunago, Atsushi Yukimoto, Masanori Abe, Yoichi Hiasa.

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
