## [Decision Letter · Decision Letter 0]

16 Nov 2020

PONE-D-20-32923

Development of a method for measuring spleen stiffness by transient elastography using a new device and ultrasound-fusion method

PLOS ONE

Dear Dr. Hiasa,

Thank you for submitting your manuscript to PLOS ONE. After careful consideration, we feel that it has merit but does not fully meet PLOS ONE’s publication criteria as it currently stands. Therefore, we invite you to submit a revised version of the manuscript that addresses the points raised by one reviewer during the review process.

We look forward to receiving your revised manuscript.

Kind regards,

Wenyu Lin, PhD

Academic Editor

PLOS ONE

Journal Requirements:

2. In your Methods section, please provide additional information about the participant recruitment method and the demographic details of your participants. Please ensure you have provided sufficient details to replicate the analyses such as: a) a description of how participants were recruited, and b) descriptions of where participants were recruited and where the research took place.

3. Please provide a sample size and power calculation in the Methods, or discuss the reasons for not performing one before study initiation.

4.Thank you for including your ethics statement: 

"The study was approved by the Ethics Committee (IRB No. jRCTs062190005). The nature of the study was explained to the patients, and written informed consent was obtained from each patient according to the principles of the Declaration of Helsinki.".   

5.Thank you for submitting the above manuscript to PLOS ONE. During our internal evaluation of the manuscript, we found significant text overlap between your submission and the following previously published works.

https://doi.org/10.1002/hep.31579 (lines 32-57)

https://doi.org/10.1111/j.1440-1746.2010.06325.x (lines 61-62)

http://docplayer.net/35808840-Mr-imaging-of-the-spleen-spectrum-of-abnormalities-1.html (lines 285-289)

https://doi.org/10.1053/j.gastro.2012.09.049 (lines 279-304)

Please revise the manuscript to rephrase the duplicated text, cite your sources, and provide details as to how the current manuscript advances on previous work. Please note that further consideration is dependent on the submission of a manuscript that addresses these concerns about the overlap in text with published work.

Reviewers' comments:

Reviewer's Responses to Questions

**Comments to the Author**

1. Is the manuscript technically sound, and do the data support the conclusions?

Reviewer #1: Yes

2. Has the statistical analysis been performed appropriately and rigorously? 

Reviewer #1: Yes

3. Have the authors made all data underlying the findings in their manuscript fully available?

Reviewer #1: Yes

4. Is the manuscript presented in an intelligible fashion and written in standard English?

Reviewer #1: Yes

5. Review Comments to the Author

Reviewer #1: I had the opportunity to review your paper “Development of a method for measuring spleen stiffness by transient elastography using a new device and ultrasound-fusion method” which I found very interesting. This paper is written clearly but there is a point that deserve attention:

Major point

1. Endoscopic findings of esophagogastric varices should be defined according to "General Rules for Recording Endoscopic Findings of Esophagogastric Varices”.

6. PLOS authors have the option to publish the peer review history of their article (what does this mean?). If published, this will include your full peer review and any attached files.

Reviewer #1: **Yes: **Hiroshi Yoshida

---

## [Author Response · Author response to Decision Letter 0]

8 Jan 2021

January 8, 2021

Dr. Wenyu Lin, PhD

Academic Editor

Dear Dr. Wenyu Lin and Zsanett Szabo

Thank you very much for providing us with the opportunity to resubmit our manuscript. We have revised the paper based on the comments of the reviewers.

We appreciate the reviewers’ input, which we believe strengthened our manuscript considerably. Similarly, we have provided a point-by-point response to each of the comments in the pages that follow. In the revised version of the manuscript, the revised sections are indicated in green font.

We reaffirm that the findings of our study have not been submitted for publication before, nor are they under consideration elsewhere.

Thank you very much for considering our revised manuscript. We look forward to your response.

Sincerely,

Yoichi Hiasa

Department of Gastroenterology and Metabology

Ehime University Graduate School of Medicine

Shitsukawa, Toon, Ehime 791-0295, Japan

Tel: +81-89-960-5308; Fax: +81-89-960-5310

E-mail: hiasa@m.ehime-u.ac.jp 

1. Thank you for including your ethics statement on the online submission form: "The study was approved by the Accredited Clinical Research Review Committee National University Corporation Ehime University Clinical Research Review Committee (IRB No. jRCTs062190005). The nature of the study was explained to the patients, and written informed consent was obtained from each patient according to the principles of the Declaration of Helsinki". To help ensure that the wording of your manuscript is suitable for publication, would you please also add this statement at the beginning of the Methods section of your manuscript file.

We appreciate the reviewers for these comments. In previous manuscripts, the following text was at the end of Study design: "The study was approved by the Accredited Clinical Research Review Committee National University Corporation Ehime University Clinical Research Review Committee (IRB No. jRCTs062190005). The nature of the study was explained to the patients, and written informed consent was obtained from each patient according to the principles of the Declaration of Helsinki ". In the revised manuscript, this statement has been changed to appear at the beginning of study design in the Methods section. (From page 9, line 89 to page 10, line 93).

2. Thank you for addressing text overlap in your submission. One sentence of overlap still remains in the second paragraph of the Discussion section:"The spleen is an intraperitoneal organ with a smooth serosal surface and is attached to the retroperitoneum by fatty ligaments that also contain its vascular supply. The splenic surfaces are described relative to their locations and are termed the diaphragmatic (phrenic) and visceral surfaces. The visceral surface is divided into an anterior or gastric ridge and a posterior or renal portion."http://docplayer.net/35808840-Mr-imaging-of-the-spleen-spectrum-of-abnormalities-1.htmlPlease ensure you cite all your sources (including your own works), and quote or rephrase any duplicated text. Further consideration is dependent on this concern being addressed.

We appreciate the reviewers for these comments. When I checked the text of the part you pointed out, I have already corrected it with the revised manuscript I submitted the other day. This time, we are changing the font color. Please confirm. (From page 36, line 304 to page 36, line 309).

December 23, 2020

Dr. Wenyu Lin, PhD

Academic Editor

Dear Dr. Wenyu Lin,

Thank you very much for providing us with the opportunity to resubmit our manuscript. We have revised the paper based on the comments of the reviewers.

We appreciate the reviewers’ input, which we believe strengthened our manuscript considerably. Similarly, we have provided a point-by-point response to each of the comments in the pages that follow. In the revised version of the manuscript, the revised sections are indicated in red font.

We reaffirm that the findings of our study have not been submitted for publication before, nor are they under consideration elsewhere.

Thank you very much for considering our revised manuscript. We look forward to your response.

Sincerely,

Yoichi Hiasa

Department of Gastroenterology and Metabology

Ehime University Graduate School of Medicine

Shitsukawa, Toon, Ehime 791-0295, Japan

Tel: +81-89-960-5308; Fax: +81-89-960-5310

E-mail: hiasa@m.ehime-u.ac.jp 

We appreciate the reviewers for these comments. Accordingly, we have checked the style requirements of PLOS ONE and modified the file names and contribution. Similarly, we have revised the font of the manuscript title to 18pt. (Page 1, lines 1-3, page 1, lines 5-7, page 2, line 12, page 2, line20).

2. In your Methods section, please provide additional information about the participant recruitment method and the demographic details of your participants. Please ensure you have provided sufficient details to replicate the analyses such as: a) a description of how participants were recruited, and b) descriptions of where participants were recruited and where the research took place.

We appreciate the reviewers for these comments. Accordingly, we have added information regarding the recruitment of participants and participant demographics.

a) Patients were admitted to the Department of Gastroenterology and Metabology, Ehime University Hospital from June 2019 to September 2019. Patients who underwent abdominal ultrasonography and liver and spleen stiffness measured using transient elastography were recruited. b) The selection criteria were: 1) 20 to 90 years of age and 2) Performance Status (ECOG) of 0 to 2. Notably, the patient’s sex was not considered in the course of selection. Abdominal ultrasonography was performed in the laboratory of the inpatient ward; on the same day, the patients underwent B-mode examination, SWE, and Fibroscan on an empty stomach more than 2 hours after eating. All participants were provided with a written explanation of the purpose, method, and details regarding handling of personal information of the research (From page 9, line 89 to page 10, line 100).

3. Please provide a sample size and power calculation in the Methods, or discuss the reasons for not performing one before study initiation.

We would like to thank the reviewer for this comment. Accordingly, calculation of the sample size and detection power has been described in the Methods section of the revised manuscript (From line 101 on page 10 to line 115 on page 12). 

4. Thank you for including your ethics statement: "The study was approved by the Ethics Committee (IRB No. jRCTs062190005). The nature of the study was explained to the patients, and written informed consent was obtained from each patient according to the principles of the Declaration of Helsinki.". Please amend your current ethics statement to include the full name of the ethics committee/institutional review board(s) that approved your specific study. Once you have amended this/these statement(s) in the Methods section of the manuscript, please add the same text to the “Ethics Statement” field of the submission form (via “Edit Submission”). For additional information about PLOS ONE ethical requirements for human subjects research, please refer to http://journals.plos.org/plosone/s/submission-guidelines#loc-human-subjects-research.

We would like to thank the reviewer for this comment. Accordingly, we have amended the Ethics Statement to include the full name of the Ethics Committee / Institutional Review Board that approved our study (From line 130 on page 13 to line 132 on page 14).

5. Thank you for submitting the above manuscript to PLOS ONE. During our internal evaluation of the manuscript, we found significant text overlap between your submission and the following previously published works.

https://doi.org/10.1002/hep.31579 (lines 32-57)

https://doi.org/10.1111/j.1440-1746.2010.06325.x (lines 61-62)

http://docplayer.net/35808840-Mr-imaging-of-the-spleen-spectrum-of-abnormalities-1.html (lines 285-289)

https://doi.org/10.1053/j.gastro.2012.09.049 (lines 279-304)

We would like to make you aware that copying extracts from previous publications, especially outside the methods section, word-for-word is unacceptable. In addition, the reproduction of text from published reports has implications for the copyright that may apply to the publications. Please revise the manuscript to rephrase the duplicated text, cite your sources, and provide details as to how the current manuscript advances on previous work. Please note that further consideration is dependent on the submission of a manuscript that addresses these concerns about the overlap in text with published work. We will carefully review your manuscript upon resubmission, so please ensure that your revision is thorough.

We would like to thank the reviewer for indicating this. Accordingly, we have revised the manuscript by rephrasing the duplicated texts (to different expressions) and quoting the sources in relevant instances.

The part of https://doi.org/10.1002/hep.31579 (lines 32-57) has been corrected (From page 3, line 24 to page 6, line 51).

The part of https://doi.org/10.1111/j.1440-1746.2010.06325.x (lines 61-62) has been corrected (Page 6, line 54 to line 55).

The part of http://docplayer.net/35808840-Mr-imaging-of-the-spleen-spectrum-of-abnormalities-1.html (lines 285-289) has been corrected Page 36, line 305 to line 310).

The part of https://doi.org/10.1053/j.gastro.2012.09.049 (lines 279-304) has been corrected (From page 35, line 300 to page 38, line 327).

6. Reviewer #1: I had the opportunity to review your paper “Development of a method for measuring spleen stiffness by transient elastography using a new device and ultrasound-fusion method” which I found very interesting. This paper is written clearly but there is a point that deserve attention:

Major point

1. Endoscopic findings of esophagogastric varices should be defined according to "General Rules for Recording Endoscopic Findings of Esophagogastric Varices”.

We appreciate the reviewer for indicating this. The endoscopic findings of the esophagogastric varices have been modified according to the "General Rules for Recording Endoscopic Findings of Esophagogastric Varices ". Although some changes have been made to the description of varices in the manuscript before this round, the endoscopists evaluated the endoscopic findings according to the " General Rules for Recording Endoscopic Findings of Esophagogastric Varices" in this study. No change was observed in the result obtained (From page 19, line 187 to page 20, line 192).

---

## [Decision Letter · Decision Letter 1]

18 Jan 2021

Development of a method for measuring spleen stiffness by transient elastography using a new device and ultrasound-fusion method

PONE-D-20-32923R1

Dear Dr. Hiasa,

We’re pleased to inform you that your manuscript has been judged scientifically suitable for publication and will be formally accepted for publication once it meets all outstanding technical requirements.

Kind regards,

Wenyu Lin, PhD

Academic Editor

PLOS ONE

Additional Editor Comments (optional):

Dear Yoichi,

Congratulations on your nice work!

Best wishes,

Wenyu

Reviewers' comments:

Reviewer's Responses to Questions

**Comments to the Author**

1. If the authors have adequately addressed your comments raised in a previous round of review and you feel that this manuscript is now acceptable for publication, you may indicate that here to bypass the “Comments to the Author” section, enter your conflict of interest statement in the “Confidential to Editor” section, and submit your "Accept" recommendation.

Reviewer #1: All comments have been addressed

2. Is the manuscript technically sound, and do the data support the conclusions?

Reviewer #1: Yes

3. Has the statistical analysis been performed appropriately and rigorously? 

Reviewer #1: Yes

4. Have the authors made all data underlying the findings in their manuscript fully available?

Reviewer #1: Yes

5. Is the manuscript presented in an intelligible fashion and written in standard English?

Reviewer #1: Yes

6. Review Comments to the Author

Reviewer #1: I had the opportunity to review your paper “Development of a method for measuring spleen stiffness by transient elastography using a new device and ultrasound-fusion method”. There is no problem to publish the manuscript.

7. PLOS authors have the option to publish the peer review history of their article (what does this mean?). If published, this will include your full peer review and any attached files.

Reviewer #1: No

---

## [Editor Report · Acceptance letter]

22 Jan 2021

PONE-D-20-32923R1 

Development of a method for measuring spleen stiffness by transient elastography using a new device and ultrasound-fusion method 

Dear Dr. Hiasa:

I'm pleased to inform you that your manuscript has been deemed suitable for publication in PLOS ONE. Congratulations! Your manuscript is now with our production department. 

Kind regards, 

on behalf of

Dr. Wenyu Lin 

Academic Editor

PLOS ONE